# Nkx2-5 defines distinct scaffold and recruitment phases during formation of the murine cardiac Purkinje fiber network

Caroline Choquet [1], Robert G. Kelly[1] & Lucile Miquerol [1]✉

The ventricular conduction system coordinates heartbeats by rapid propagation of electrical activity through the Purkinje fiber (PF) network. PFs share common progenitors with contractile cardiomyocytes, yet the mechanisms of segregation and network morphogenesis are poorly understood. Here, we apply genetic fate mapping and temporal clonal analysis to identify murine cardiomyocytes committed to the PF lineage as early as E7.5. We find that a polyclonal PF network emerges by progressive recruitment of conductive precursors to this scaffold from a pool of bipotent progenitors. At late fetal stages, the segregation of conductive cells increases during a phase of rapid recruitment to build the definitive PF network through a non-cell autonomous mechanism. We also show that PF differentiation is impaired in Nkx2-5 haploinsufficient embryos leading to failure to extend the scaffold. In particular, late fetal recruitment fails, resulting in PF hypoplasia and persistence of bipotent progenitors. Our results identify how transcription factor dosage regulates cell fate divergence during distinct phases of PF network morphogenesis.

[1] Aix-Marseille Université, CNRS UMR 7288, Developmental Biology Institute of Marseille, Campus de Luminy Case 907, 13288 Marseille, Cedex 9, France.
✉email: lucile.miquerol@univ-amu.fr

The cardiac conduction system (CCS) is essential to initiate and propagate electrical activity throughout the entire heart[1,2]. Efficient heartbeats require perfect coordination between conduction and contraction driven by specialized conductive and contractile cardiomyocytes, respectively. In the ventricles, the terminal conduction system consists of Purkinje fiber cells (PFs) characterized by fast conduction properties necessary to activate the ventricular myocardium from the apex to the base and ensure the efficient expulsion of the blood. PFs are organized in a complex network of fascicles forming ellipsoidal structures[3]. Despite their functional importance and identification of PFs as a major trigger of ventricular arrhythmias[4], it is still unclear how the complex PF network forms during ventricular morphogenesis[5–8].

During ventricular development, trabecular myocardium expressing the gap junction protein Connexin40 (Cx40; Gja5) constitutes the primary fast-conducting pathway[9]. An important finding came from clonal studies in chick, showing that PFs shared a common myogenic origin with contractile cardiomyocytes of the working myocardium[10]. More recently, using genetic tracing and retrospective clonal analysis, it has been demonstrated that murine ventricular trabeculae contain PF progenitor cells. Moreover, the ventricular conduction system (VCS) displays a biphasic development involving lineage restriction to a conductive phenotype followed by limited proliferative outgrowth. Lineage restriction between trabecular cardiomyocytes and PFs is largely complete by E16.5[11,12]. However, key questions concern when and how trabecular cells diverge towards the conductive or contractile lineage during development. To identify the timing of lineage segregation of contractile and conductive cardiomyocytes within the trabecular progenitor cell population, an in vivo temporal clonal analysis is required. Clonal analysis, consisting of the genetic labeling of an individual cell at a defined developmental stage and studying its progeny at a later stage, is a powerful approach to assess cell fate and cell behavior within a heterogeneous progenitor population.

The transcription factor NKX2-5 is highly conserved across evolution in vertebrates and one of the earliest markers of the developing heart[13]. In the human population, NKX2-5 mutations represent around 4% of congenital heart diseases (CHDs) with a high prevalence of conduction disturbances and arrhythmias[14]. $Nkx2-5^{+/-}$ mice reproduce the cardiac phenotype observed in patients, including atrial septal defects and atrioventricular (AV) conduction defects associated with hypoplastic development of the conduction system[15–17]. While single PF cells from Nkx2-5 haploinsufficient mice display normal electrophysiological properties, ventricular conduction defects such as prolonged QRS duration, conduction blocks, and inducible ventricular arrhythmias have been correlated with a reduced PF network lacking ellipsoidal structures[16–18]. Therefore, detailed understanding of how the PF network forms during development is required. Moreover, patients carrying NKX2-5 mutations display noncompaction cardiomyopathy[19] and Nkx2-5 expression is required during trabecular development to prevent hypertrabeculation[20,21]. In this study, by performing temporal clonal analysis of cardiac progenitors, we identify distinct phases of PF network morphogenesis: an initial bipotent-progenitor-dependent scaffolding phase and a later conductive progenitor recruitment phase. Extending our analysis to Nkx2-5 heterozygous mutant mice we define the temporal requirement for Nkx2-5 during the segregation of trabecular cells towards the conductive lineage and gain mechanistic insight into the etiology of PF hypoplasia.

## Results

**Nkx2.5 dosage selectively affects PF cell fate**. To establish the lineage contribution of trabecular cells to the VCS during ventricular development, we performed genetic fate mapping of Cx40+ cells. Cx40 expression is first observed in atrial and ventricular cardiomyocytes at embryonic day (E) E9.5[22]. Between E10.5 and E14.5, Cx40 is strongly expressed in ventricular trabeculae and restricted to conductive cells of the VCS at birth[12]. Cx40-CreERT2[23] mice were crossed with Rosa26-YFP conditional lineage reporter mice and Cre recombination was activated by tamoxifen injections at different embryonic days, prior to analysis at postnatal day 21 (P21; Supplementary Fig. 1). Cells expressing Cx40 in the embryonic (E10.5) and fetal (E14.5) heart actively contribute to both the left ventricular free wall and VCS. In contrast, after tamoxifen injection at P1, Cx40-derived cell numbers are reduced and restricted to the VCS. Consistent with previous reports, this shows the progressive fate restriction of Cx40-positive trabecular cells to the VCS lineage[24].

In order to investigate the mechanisms underlying PF hypolasia in Nkx2-5 haploinsufficient hearts, we performed similar inducible genetic tracing in Nkx2-5 heterozygous mice. Cx40-trabecular cells were labeled at E14.5 and their contribution to the VCS assayed by Cx40-RFP (PF) and YFP (Trabecular-derived) expression at P7 (Fig. 1a–c). Consistent with PF hypoplasia, we observed a reduced number of RFP + PF in $Nkx2-5^{+/-}$ hearts (Fig. 1d), however, the total number of trabecular-derived cells is similar in $Nkx2-5^{+/-}$ and control hearts (Fig. 1e, Supplementary Fig. 1). Indeed, the contribution of trabecular cells to the VCS is highly reduced with only 6% of YFP+ cells expressing RFP in $Nkx2-5^{+/-}$ hearts compared to 20% in the wild-type situation (Fig. 1f). Fetal Cx40+ trabecular cells thus predominantly contribute to working myocardium in Nkx2-5 haploinsufficient hearts, demonstrating that the acquisition of PF cell fate is selectively and severely affected (Fig. 1g).

**Trabecular cells segregate into the PF lineage in two phases**. To determine whether trabecular cells exhibit bipotency at the cellular level, and when the segregation of conductive and contractile cell fate choice occurs, we performed clonal analysis using the multicolor Rosa26-Confetti reporter mouse line at different timepoints of development. We used Cx40-CreERT2 mice to genetically mark trabecular cardiomyocytes with a low dose of 4OH-tamoxifen. This generates hearts containing distinct clusters derived from different individual cells stochastically labeled by one of the four colors on recombination of the confetti cassette (Fig. 2a, b; Supplementary Fig. 2a). The absence of clusters containing multicolor cells is a strong argument that cells within each cluster can be considered to be clonally related[25]. The frequency and distribution of clusters observed in control and $Nkx2-5^{+/-}$ hearts were identical, indicating that the probability of generating Cx40-derived clones is not affected by reduced levels of Nkx2-5 (Table 1; Supplementary Fig. 2b). Immunofluorescence against Contactin 2 (CNTN2), identifying mature PF cells, was used to distinguish three types of clusters. Conductive clones are composed of 100% CNTN2-positive cells and are likely to result from labeling of a lineage-restricted conductive precursor cell. Clones in which only a sub-population of labeled cells are CNTN-2-positive were classified as mixed and originate from a bipotent progenitor. Finally, non-conductive clones with no CNTN2-positive cells originate from recombination in a working cardiomyocyte precursor cell (Fig. 2c).

Clonal labeling of Cx40+ cells at E9.5 produced a collection of conductive, mixed, and non-conductive clones, indicating the heterogeneity of Cx40+ progenitors present at this timepoint. Unexpectedly, 39% of these unicolor clones contained exclusively CNTN2-positive conductive cells, indicating that a third of Cx40+ trabecular cells are already committed to a PF fate at E9.5 (Fig. 2d). A similar percentage was observed in both $Nkx2-5^{+/-}$

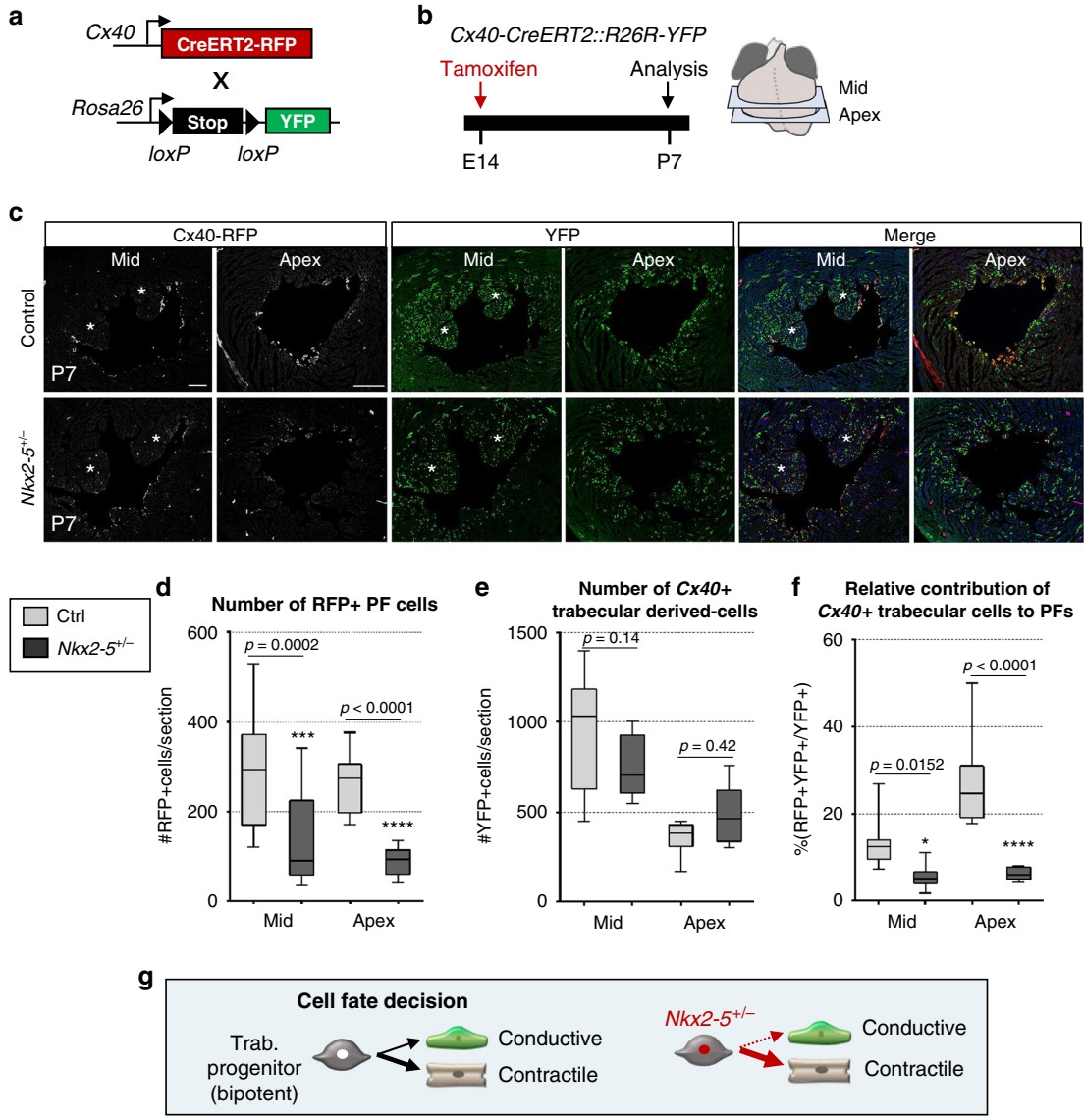

**Fig. 1 Defective contribution of *Cx40*+ derived trabecular cells to the postnatal PF in *Nkx2-5*+/− mice. a** Scheme of the strategy used for genetic lineage tracing of *Cx40*+ cells. *R26R-YFP* reporter mouse line are crossed with tamoxifen inducible *Cx40-CreERT2* mice. Injection of tamoxifen induces the recombination of loxP sites resulting in the expression of the fluorescent protein YFP in *Cx40*+ cells. **b** Tamoxifen is injected at E14 and *Cx40*+ derivative cells are analyzed at P7. **c** Immunostaining for YFP and RFP on *Cx40-CreERT2-RFP::R26R-YFP* control (ctrl) or *Nkx2-5*+/− heart transverse sections at P7, shows distribution of *Cx40*+ derivative cells (YFP+) and their fate into PFs (YFP+RFP+). Asterisks represent papillary muscles; scale bars = 200 μm. (*n* = 4 Ctrl and *Nkx2-5*+/−). **d**–**f** Graphs of quantifications from **c** images. The total number PF cells (RFP+) per section **d**, the total number of trabecular-derived cells (YFP+) **e**, and the relative contribution of trabecular cells to PFs ((YFP+RFP+)/YFP+) **f** are quantified in mid and apical sections. Box plots show the median, the 25th and 75th percentile, and the whiskers denote the minimum and maximum values, respectively. Mean of 3 sections/heart; (*n* = 4 Ctrl and *Nkx2-5*+/−). *P* values are derived ordinary one-way ANOVA test, *P < 0.05; ***P < 0.001. **g** Scheme illustrating the differentiation of embryonic trabecular progenitors into conductive and contractile cardiomyocytes. In *Nkx2-5*+/− mice, the cell fate decision into conductive cardiomyocytes is disturbed and haploinsufficient trabecular progenitors mostly give rise to contractile cardiomyocytes.

and control littermates, indicating that *Nkx2-5* haploinsufficiency does not impact on early commitment to the PF lineage (Fig. 2e). Induced labeling at later stages of development demonstrates the progressive loss of bipotency of *Cx40*+ trabecular cells, as shown by a reduction in the number of mixed clones over time (Fig. 2f). Conversely the proportion of conductive clones increases with time, consistent with progressive restriction of fate and *Cx40* expression to the conductive lineage. Independent of their date of birth or whether they are generated in wildtype or *Nkx2-5* heterozygous mutant hearts, conductive clones always contain a small number of cells (<20), consistent with loss of proliferation on segregation to the conductive lineage (Supplementary Fig. 3).

This implies that growth of the VCS occurs by progressive recruitment of committed progenitors as proposed by the ingrowth model[5,26]. Tracing the kinetics of apparition of the conductive clones defines an early phase of slow recruitment until E14.5 followed by a phase of rapid recruitment at late fetal stages (Fig. 2f). In *Nkx2-5*+/− mice, the progressive recruitment of committed progenitors to the VCS is severely affected, preventing their differentiation into conductive cells, particularly during the phase of rapid recruitment as shown by the high proportion of non-conductive clones labeled at E18.5. In contrast to the wildtype situation, mixed clones are observed in *Nkx2-5*+/− hearts after labeling at E18.5, identifying that *Cx40*+ bipotent

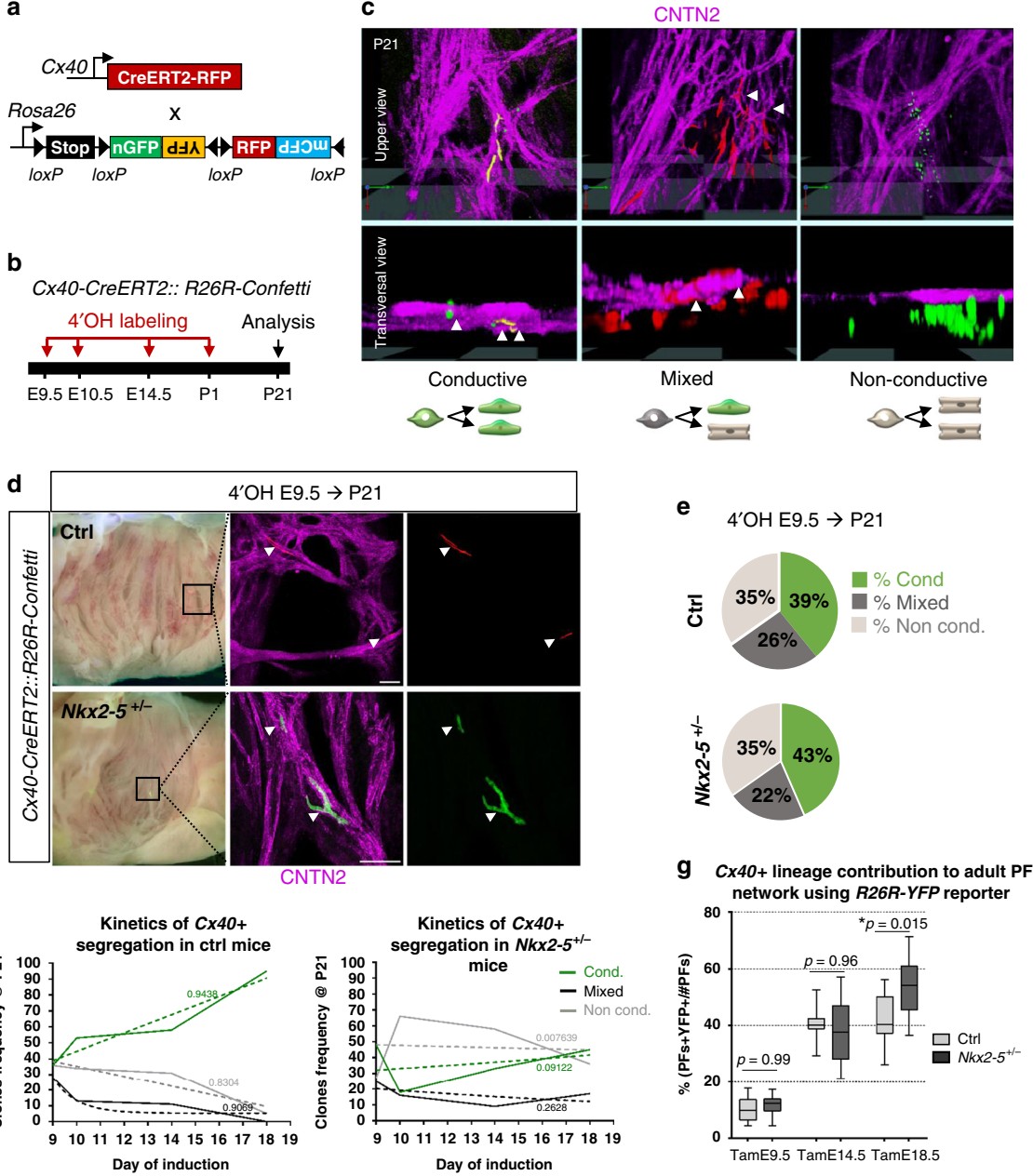

**Fig. 2 Temporal clonal analysis of single *Cx40+* trabecular cells during embryonic stages. a** Scheme of the strategy used for clonal genetic tracing of *Cx40+* cells. *R26R-Confetti* multicolor reporter mice are crossed with tamoxifen-inducible *Cx40-CreERT2* mice. Four alternative recombination (nGFP, YFP, RFP, and mCFP) are possible. **b** Low doses of 4-hydroxytamoxifen (4'OH-Tam) are injected at different time points of development to induce low-frequency recombination. Analyses of independent unicolor clones are made at P21. **c** 3D reconstructions of clones imaged at the subendocardial surface of the left ventricle. Whole-mount immunostaining for Contactin-2 (CNTN2) is used to distinguish conductive, mixed and non-conductive clones. Arrowheads indicate cells positive for CNTN2. Schemes illustrate the lineage origin of each clone. **d** Confocal images of whole-mount CNTN2-immunostaining of 3-week-old *Cx40-CreERT2::R26R-Confetti* opened-left ventricle induced at E9.5 from control (Ctrl) and *Nkx2-5+/−* hearts. Conductive clones are indicated by arrowheads. Scale bars = 100 μm. **e** Percentages of conductive, mixed and non conductive clones induced at E9.5 were quantified in P21 control (Ctrl) and *Nkx2-5+/−* left ventricles. **f** Clone frequency evolution upon time-course inductions quantified in P21 control (*Ctrl*) and *Nkx2-5+/−* mice. The progressive decrease of mixed clones over time illustrates the kinetics of *Cx40+* trabecular progenitors segregation. Dashed lines: non-linear regression curves with *R* square values. **g** Graphs of the percentage of labeled adult PFs according to *Cx40+* lineage traced by Tam injections at E9.5, E14.5, or E18.5. Box plots show the median, the 25th and 75th percentile, and the whiskers denote the minimum and maximum values, respectively. Quantifications were performed on *Cx40-CreERT2::R26R-YFP* control (ctrl) or *Nkx2-5+/−* heart transverse sections. Mean of 3 sections/heart (Tam E14.5; Tam E18.5); 4 sections/heart (TamE9.5); (*n* = 3 TamE9.5 Ctrl, TamE18.5 Ctrl and *Nkx2-5+/−*; *n* = 4 TamE9.5 *Nkx2-5+/−*, TamE14.5 Ctrl and *Nkx2-5+/−*). *P* values are derived ordinary one-way ANOVA test (Tukey test), *$P < 0.05$.

progenitor cells persist abnormally in *Nkx2-5+/−* hearts at birth. Together these results indicate a requirement of maximal levels of *Nkx2-5* for progressive recruitment to the VCS during fetal development.

To study if the hypoplasia of the PF network in Nkx2-5 heterozygotes mice results from a disturbance in the timing of these two phases of recruitment, we performed temporal genetic fate mapping and quantified the relative proportion of trabecular

**Table 1 Quantitative analysis of _Cx40+_-derived clones.**

| | 4'OH E9.5 | | 4'OH E10.5 | | 4'OH E14.5 | | 4'OH E18.5 | |
|---|---|---|---|---|---|---|---|---|
| | Ctrl | _Nkx2-5_$^{+/-}$ | Ctrl | _Nkx2-5_$^{+/-}$ | Ctrl | _Nkx2-5_$^{+/-}$ | Ctrl | _Nkx2-5_$^{+/-}$ |
| Total hearts | 10 | 15 | 16 | 10 | 8 | 9 | 6 | 4 |
| Total clones | 36 | 48 | 64 | 50 | 118 | 128 | 105 | 58 |
| Frequency/heart | 3.6 | 3.2 | 4 | 5 | 14.75 | 14.2 | 17.5 | 14.5 |
| % GFP+ | 39.1 | 30.4 | 37.5 | 44 | 39.3 | 37.5 | 34.3 | 37.9 |
| % CFP+ | 4.3 | 4.3 | 1.6 | 2 | 0.85 | 0 | 0.95 | 0 |
| % RFP+ | 43.5 | 52.2 | 42.2 | 40 | 38.5 | 41.4 | 45.7 | 43.1 |
| % YFP+ | 13.0 | 13.0 | 18.75 | 14 | 21.4 | 21.1 | 19.0 | 19.0 |
| % Cond. clones | 36.1 | 47.9 | 53.1 | 18 | 57.6 | 32.8 | 95.2 | 44.8 |
| % Mixed clones | 27.8 | 25 | 12.5 | 16 | 11.0 | 9.4 | 0 | 17.2 |
| % Non-cond. clones | 36.1 | 27.1 | 34.4 | 66 | 37 | 57.8 | 5 | 36.2 |
| Size cond. clones | 5.4 | 5.1 | 4.5 | 3.9 | 2.6 | 2.5 | 3.18 | 1.9 |
| Size mixed clones | 27.7 | 13.3 | 31.1 | 9.25 | 6.5 | 7.25 | 0 | 6 |
| Size non-cond. clones | 16.9 | 27.7 | 31.6 | 16.1 | 6.5 | 6.7 | 4.8 | 3.0 |

Table represents the _n_ values of hearts and clones analyzed in this study, using Cx40-CreERT2::R26R-Confetti mouse line. Only clones observed in the subendocardial surface of the inter-ventricular septum within the left ventricle are analyzed. Percentages of colors and type of clones are represented. The size of clones are measured by quantifying the number of cells per clone.

cells giving rise to PFs. A similar proportion of trabecular cells contributing to PFs at E9.5 and E14.5 was observed in control and _Nkx2-5_$^{+/-}$ hearts, reinforcing the idea that the kinetics of the early phase of recruitment is unchanged by a reduced level of Nkx2-5 (Fig. 2g and Supplementary Fig. 4a–c). In contrast, when labeled at E18.5, trabecular cells contribute to a larger proportion of the _Nkx2-5_$^{+/-}$ PF (54%) at P7 compared to control hearts (42%) (Fig. 2g). This supports the absence of rapid recruitment of new conductive cells in _Nkx2-5_$^{+/-}$ hearts at late fetal stages. Moreover, this shows that at E9.5, the onset of trabeculation, labeled trabecular cells only give rise to a small proportion of PFs (10%) while this proportion increases on labeling at E14.5 (35%). Early Cx40+ trabecular cells thus do not represent the totality of VCS progenitors and Cx40-negative progenitor cells contribute to later growth of the PF network.

**Segregation of PF progenitor cells in the linear heart tube**. To study the contribution of earlier progenitors to the VCS, the smooth muscle actin _SMA-CreERT2_ mouse line was used to label differentiated cardiomyocytes from the cardiac crescent stages (E7.5), prior to the onset of _Cx40_ expression[27]. Genetic tracing using Rosa-YFP reporter mice shows that early cardiomyocytes contribute to PFs in equivalent proportions in control and _Nkx2-5_ heterozygous mutant embryos (Supplementary Fig. 4d). We then examined the cellular behavior of early labeled cardiomyocyte clones during segregation of conductive and contractile fates using _Rosa-confetti_ mice and a low dose of 4OH-tamoxifen (Fig. 3a, b). As SMA is expressed in all cardiomyocytes at this stage, we observed a large number of non-conductive clones in 3-weeks-old whole hearts from control and _Nkx2-5_$^{+/-}$ mice (Supplementary Fig. 5) and focused our study on conductive and mixed clones. Consistent with early segregation of cardiac progenitors towards the PF lineage, conductive clones were observed after labeling at both E8.5 and E7.5 (Supplementary Fig. 6). Such conductive clones were observed in both control and _Nkx2-5_ heterozygous hearts (Fig. 3c). Together, these results demonstrate that the early segregation of the PF lineage within SMA-positive cells starts at or before the onset of heart tube formation, before trabeculation begins. Moreover, consistent with our observations using the _Cx40-CreERT2_ line, this early segregation is unaffected by _Nkx2-5_ haploinsufficiency[28].

Tracing the relative proportion of mixed clones induced at different embryonic days E7.75, E8.5, E9.5, and E10.5, identified a similar decrease of bipotent progenitors with time both in control

and _Nkx2-5_$^{+/-}$ hearts (Fig. 3d). This illustrates the progressive lineage restriction of cardiac progenitor cells to working or conductive fate during the early recruitment phase. However, a higher number of mixed clones were observed in _Nkx2-5_ heterozygous hearts at each time point, suggesting that a progressively larger number of SMA-positive cardiac progenitors fail to segregate into the conductive lineage.

A detailed analysis of the clone size distribution induced by SMA-CreERT2 at E8.5 identified that mixed clones are heterogeneous in size and that large mixed clones (>80) are absent in _Nkx2-5_$^{+/-}$ hearts (Fig. 3e). In control hearts, the presence of large mixed clones is consistent with the high proliferative capacity of bipotent progenitors while the conductive compartment remains small and of comparable size between control and _Nkx2-5_$^{+/-}$ hearts (Supplementary Fig. 3, Table 2). These data show that a reduced level of _Nkx2-5_ affects the cellular behavior of bipotent progenitors originating from early differentiating cardiomyocytes. Two hypotheses may explain this discrepancy: either a reduced proliferative capacity of these bipotent progenitors or an absence of recruitment of proliferative progenitors to the VCS. Together, these data suggest that the early (E8.5) heart contains a population of progenitor cells that do not express Cx40 and will later contribute to trabeculae and the VCS. The subsequent development of this population of progenitor cells is particularly affected by a reduced level of Nkx2-5 impacting their later contribution to the PF network.

**Reduced Nkx2-5 dosage impairs recruitment of PF progenitors**. To further investigate the mechanism underlying the lack of recruitment of cells derived from early progenitors to the VCS, we studied the contribution of all cardiac progenitors by mosaic tracing analysis using _Mesp1-Cre_ and _Rosa-confetti_ mice (Fig. 4a, b). _Mesp1_ is expressed during a short time window during gastrulation, and the _Mesp1-Cre_ allele induces spatio-temporally restricted Cre activation in mesodermal progenitors, generating clones born between E6.5 and E7.5[25]. The overall distribution of _Mesp1-Cre_-activated confetti (RFP, YFP, CFP, GFP) clusters is indistinguishable in hearts from control and _Nkx2-5_$^{+/-}$ mice (Fig. 4c; Supplementary Fig. 5). However, while in control hearts, the PF network composed of multiple fascicles forming ellipsoidal structures is dense and complex, in _Nkx2-5_$^{+/-}$ hearts, the network is hypoplastic with less ellipses and thinner fascicles (Fig. 4c). However, each fascicle is composed of multicolor _Mesp1_ lineage labeled cells in both wildtype and

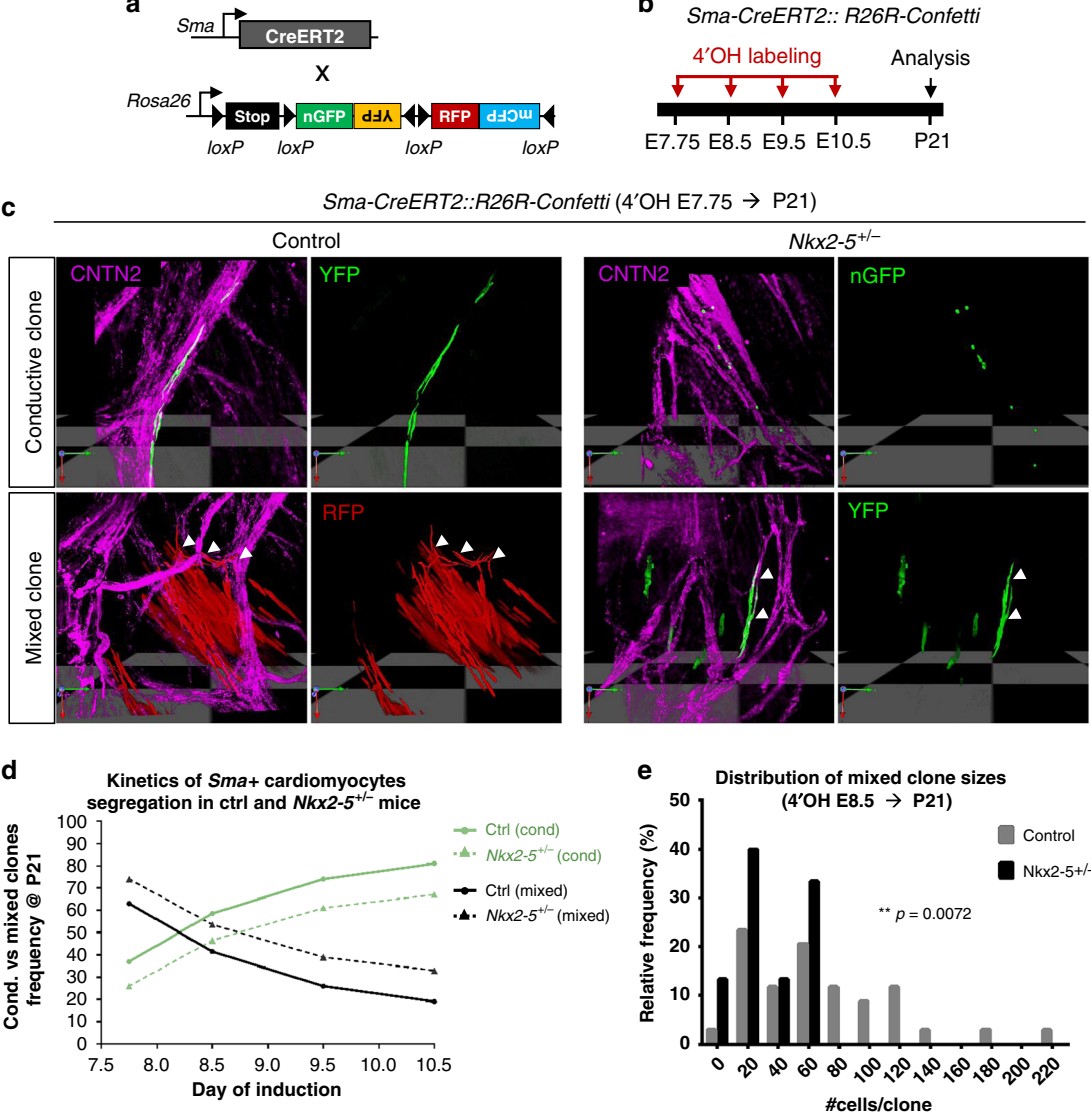

**Fig. 3 Temporal clonal analysis of single *Sma+* cardiomyocytes during embryonic stages. a** Scheme illustrating the genetic clonal tracing strategy using *R26R-Confetti* multicolor reporter mouse line crossed with tamoxifen-inducible *Sma-CreERT2* mice. **b** Time course of 4-hydroxytamoxifen (4'OH-Tam) injections and analyses of independent unicolor clones represented on a time scale. Low doses of 4'OH induce low-frequency recombination. **c** 3D reconstructions of conductive and mixed clones induced at E7.75 and observed within *Sma-CreERT2::R26R-Confetti* opened-left ventricles at P21 of control (Ctrl) and *Nkx2-5+/−* left ventricle. Whole-mount immunostaining for Contactin-2 (CNTN2) is used to distinguish conductive cells (CNTN2+) as indicate by arrowheads. **d** Conductive versus mixed clones frequency evolution upon time-course inductions quantified in P21 control (*Ctrl*) and *Nkx2-5+/−* left ventricle. The progressive decrease of mixed clones frequency over time illustrates the kinetics of *Sma+* cardiac progenitors segregation. **e** Diagram of the relative frequency of mixed clone sizes induced at E8.5 in P21 control and *Nkx2-5+/−* left ventricles. *P* value is derived from non-parametric Mann–Whitney *t*-test, **$p < 0.01$.

*Nkx2-5+/−* hearts (Fig. 4c). *Mesp1*-labeled cells within the VCS were highlighted to show the polyclonal development of the PF network composed of a tangle of multicolor fascicles supporting the ingrowth model (Fig. 4d). Closer examination of these images showed that the complex PF network is composed of both small and large monochromatic clusters in control hearts while only small clusters contribute to the hypoplastic network in *Nkx2-5+/−* hearts. The presence of large monochromatic clusters might reflect a clonal dominance and is consistent with the relative small number of progenitors giving rise to the myocardium in mice (~230) or zebrafish (~8) hearts[29,30]. The absence of large monochromatic clusters is consistent with the lack of recruitment of PFs from a pool of proliferative progenitors to the PF network in *Nkx2-5* haploinsufficient mice.

Next, we quantified the contribution of *Mesp1*-derived cells to the VCS on sections and could not detect a difference in the overall proportion of confetti-labeled cells between control and Nkx2-5 heterozygous mutant hearts (Supplementary Fig. 4f, g). However, comparing the ratio of the four possible single-color clusters in the two cases, we observed a majority of RFP-labeled and YFP-labeled cells in control hearts, while the proportion is almost equally distributed for RFP, YFP, and CFP in the mutant (Supplementary Fig. 4h). A lower proportion of GFP+ cells in both control and mutant hearts is explained by its nuclear localization which make these cells invisible when the nucleus is absent from the section and the fact that this color is usually less represented than the others on recombination of the Confetti cassette[31]. However, control and *Nkx2-5+/−* hearts show a similar distribution of individual confetti

**Table 2 Quantitative analysis of Sma+-derived-clones.**

| | 4′OH E7.75 | | 4′OH E8.75 | | 4′OH E9.5 | | 4′OH E10.5 | |
|---|---|---|---|---|---|---|---|---|
| | Ctrl | Nkx2-5$^{+/-}$ | Ctrl | Nkx2-5$^{+/-}$ | Ctrl | Nkx2-5$^{+/-}$ | Ctrl | Nkx2-5$^{+/-}$ |
| Total hearts | 4 | 4 | 13 | 5 | 20 | 14 | 5 | 8 |
| Total clones | 19 | 19 | 82 | 28 | 179 | 64 | 37 | 48 |
| Frequency/LV | 4.75 | 4.75 | 6.3 | 5.6 | 9.0 | 4.6 | 7.4 | 6.0 |
| % cond. clones* | 37 | 26 | 58.5 | 46.4 | 73.7 | 60.9 | 81.1 | 66.7 |
| % mixed clones* | 63 | 74 | 41.5 | 53.6 | 26.3 | 39.1 | 18.9 | 33.3 |
| Size cond. clones | 15.7 | 10.5 | 9.4 | 5.4 | 10.2 | 10.4 | 7.0 | 5.8 |
| Size mixed clones | 156.25 | 96.1 | 68.6 | 32.7 | 47.5 | 39.8 | 72.4 | 31.9 |
| Size cond. in mixed | 20.4 | 13.1 | 24.1 | 7.5 | 14.1 | 10.7 | 7.3 | 4.5 |

Table represents the n values of hearts and clones analyzed in this study, using Sma-CreERT2::R26R-Confetti mouse line. Asterisks shows percentages of conductive clones versus mixed clones. Non-conductive clones are excluded from the analysis.

colors in left ventricular myocardium, consistent with the whole-mount analysis (Supplementary Fig. 4i). In control hearts, the distribution of the proportion of each color from individual hearts in PF versus left ventricular myocardium follows a linear regression with a correlation coefficient close to 1 ($r^2 = 0.96 \pm 0.03$) demonstrating a proportional growth of these two compartments (Fig. 4e). These data fit well with recent evidence for the existence of a hybrid zone, indicating that the ventricular myocardium grows by proliferation of compact myocardium invading up to the subendocardial zone[32]. In contrast, the distribution of each color to PF and working myocardial clusters is not correlated in Nkx2-5$^{+/-}$ hearts ($r^2 = 0.32 \pm 0.27$) suggesting that the PF network grows independently from the working myocardium or that there is a limited contribution of the hybrid zone to the PF (Fig. 4f). These discrepancies may result from differences in the cell behavior of progenitor cells, such as their rate of proliferation. Indeed, inducible lineage tracing analysis of Mesp1 progenitors has identified a more proliferative subpopulation of cardiac progenitors induced at E6.75[29]. These results suggest that while PFs arise globally from the same progenitors (Mesp1 and SMA), the network is formed by different subpopulations of progenitors. Together, these data reinforce and extend our previous hypothesis suggesting that the PF network develops in two phases[18]. During the early phase of recruitment, committed conductive cells arise from poorly proliferative progenitor cells and give rise to small clusters that provide a scaffold for a polyclonal PF network, then at late fetal stages, rapid recruitment of newly committed cells participates in the building of a more complex network indicated by the contribution of large monochromatic clusters (Fig. 5). In Nkx2-5$^{+/-}$ hearts, this last recruitment step fails and the PF network remains in its early state.

**Nkx2-5 dosage regulates morphogenesis of the PF network.** To investigate whether PF hypoplasia in Nkx2-5 heterozygous mice may also result from a defect in maintenance of a conduction phenotype after birth, we carried out a genetic tracing analysis of Cx40-derived trabecular cells at birth (Fig. 6a). After E18.5 labeling, the majority of YFP+ cells were conductive in control hearts, while in Nkx2-5$^{+/-}$ hearts a large number of YFP+ cells were CNTN-2 negative (Fig. 6b, Supplementary Fig. 4c). High magnification of control hearts identified a well-organized network of YFP+/CNTN-2+ PFs forming ellipsoidal structures while YFP+/CNTN-2− cells in Nkx2-5 heterozygous hearts exhibited a parallel alignment, and presented a rectangular cell shape, which is a typical morphology of working cardiomyocytes (Fig. 6b″) in comparison with the thin and elongated shape of PF (Fig. 6a″ and Supplementary Fig. 6). These results strongly suggest that Nkx2-5 haploinsufficient Cx40-positive cells are maintained at birth as shown by the clonal analysis but fail to

differentiate into conductive cells. In order to study the temporal effect of Nkx2-5 deletion, we performed genetic tracing of Cx40+ cells in mice in which one allele of Nkx2-5 was deleted upon tamoxifen injection at E10.5 or E18.5 (Fig. 6c, d). In Nkx2-5-floxed/+ mice upon E18.5 deletion, YFP+ cells displayed a comparable cell shape and organization to that observed in control mice, indicating that reduction of Nkx2-5 expression after birth does not affect the differentiation of Cx40+ cells and that formation of the PF network is maintained. These data demonstrate that morphogenesis of the PF network occurs before birth and requires a maximal level of Nkx2-5 expression in embryonic cardiomyocytes. The genetic lineage of Cx40-derived cells at E10.5 shows a large proportion of lineage-negative (YFP−) PF in control hearts, confirming the active recruitment of new PF at late fetal stages from cells not expressing Cx40 at the time of induction (Fig. 6e, Supplementary Fig. 7). In Nkx2-5-floxed/+ mice deleted at E10.5 using Cx40-Cre, the PF network is hypoplastic and most of the YFP+ cells are not recruited. Moreover, while Nkx2-5 is partially deleted as depicted by the YFP lineage, wild-type cells are unable to rescue the PF hypoplasia phenotype suggesting that massive late stage PF recruitment occurs through a non-cell autonomous mechanism.

**Discussion**
In this study, we exploit the power of mosaic genetic tracing and temporal clonal analyses using inducible Cre lines to dissect the sequential events occurring during the development of the PF network. We demonstrate that Cx40+ trabeculae constitute a heterogeneous pool of progenitors that includes cells committed to the conductive lineage as early as E9.5. Moreover, a subpopulation of early cardiac progenitors is already fated to the PF lineage at the onset of the heart tube formation (E7.75), prior to Cx40 expression. This unexpected early segregation of cardiac progenitor cells to a PF fate is unaffected by Nkx2-5 haploinsufficiency. Recently, it has been shown that central components of the CCS are already committed before heart tube formation[28,33] and are derived from early cardiomyocytes expressing Tbx3[34]. However, it was unknown whether cells of the murine PF network, that grows at least in part through later recruitment to a PF fate, were also committed at such early stages. Our data demonstrate that peripheral VCS precursors of the PF network, like central VCS components, emerge from populations of cardiac progenitor cells that are specified but remain electrically inactive during the initial stages of heart tube development, as has been shown for pacemaker progenitor cells in avian embryos[33]. More recently Ren et al. show that these latter specialized cardiomyocytes originate from outlying Nkx2.5+ mesodermal progenitors that downregulate Nkx2-5 prior to differentiation[35]. This study highlights the different requirements

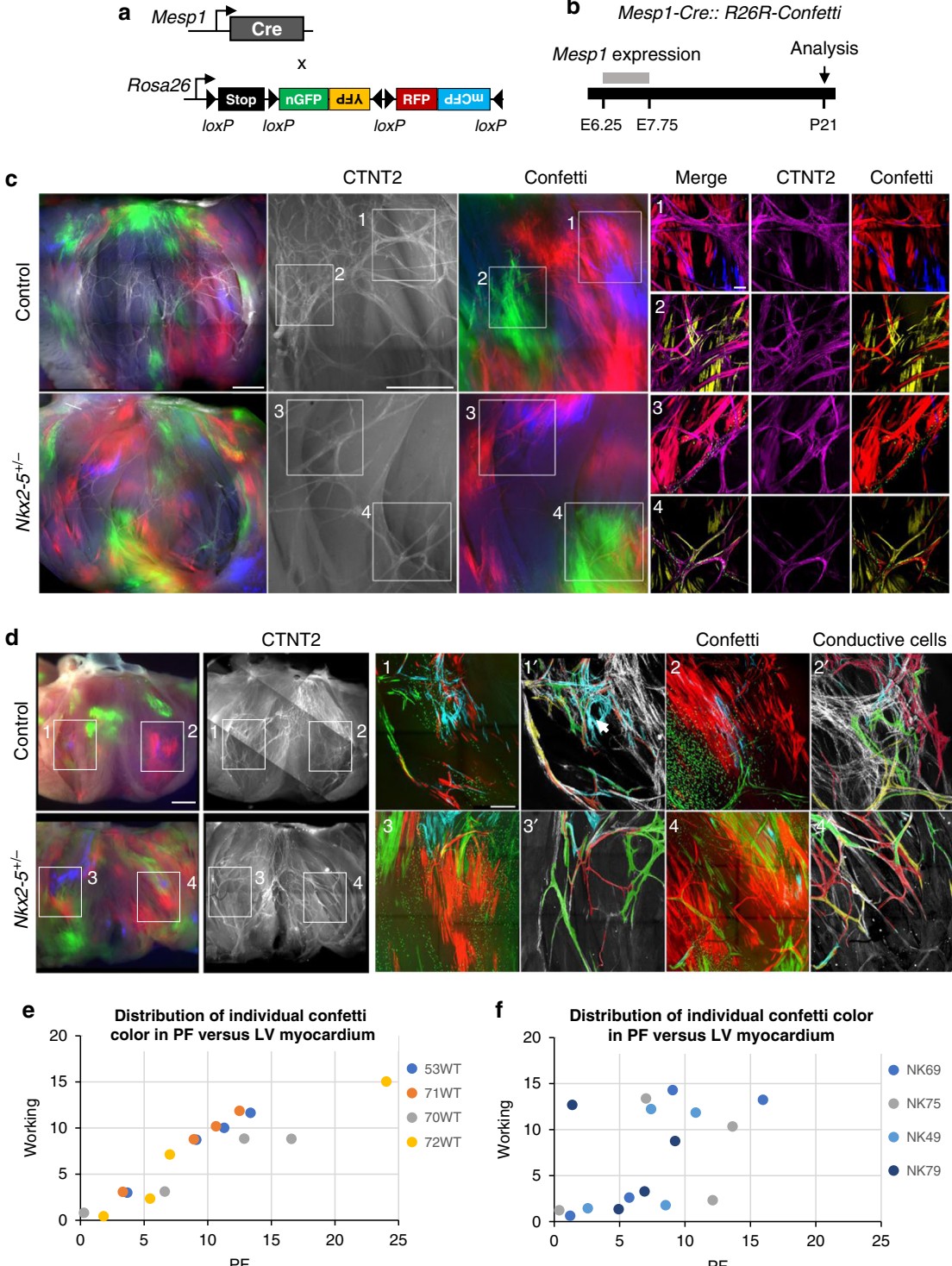

**Fig. 4 Mosaic tracing analysis of *Mesp1*+ early cardiac progenitors. a** Scheme illustrating the genetic tracing strategy using *R26R-Confetti* multicolor reporter mouse line crossed with non-inducible *Mesp1-Cre* mice. **b** Recombination of the confetti allele occurs during the short time-window of *Mesp1* expression, between E6.25 and E7.75. Mosaic tracing analysis in control and *Nkx2-5+/−* mice is performed at P21. **c** Whole-mount fluorescence views of *Mesp1-Cre::R26R-Confetti* opened-left ventricles at P21. Immunostaining for Contactin-2 (CNTN2) is used to label the mature PF network. Small panels (1–4) show high magnification confocal images of different PF network regions. Scale bars = 1 mm and 100 μm in insets. **d** Small panels (1–4) show high magnification confocal images of the PF network. Panels (1′–4′) are image reconstructions in which conductive only confetti+ cells are manually drawn. Non-conductive confetti+ cells are not represented. In control hearts, conductive confetti+ cells form either small or large monochromatic clusters. Large clusters (arrows) contribute to complex parts of the PF network. In *Nkx2-5+/−* hearts, only small clusters contribute to the hypoplastic network. Scale bars = 1 mm and 200 μm in insets. **e** and **f** Dot plots of individual *Mesp1*+ confetti colors showing percentage contribution to the working myocardium versus PF network in control (WT) or *Nkx2-5+/−* (NK) hearts.

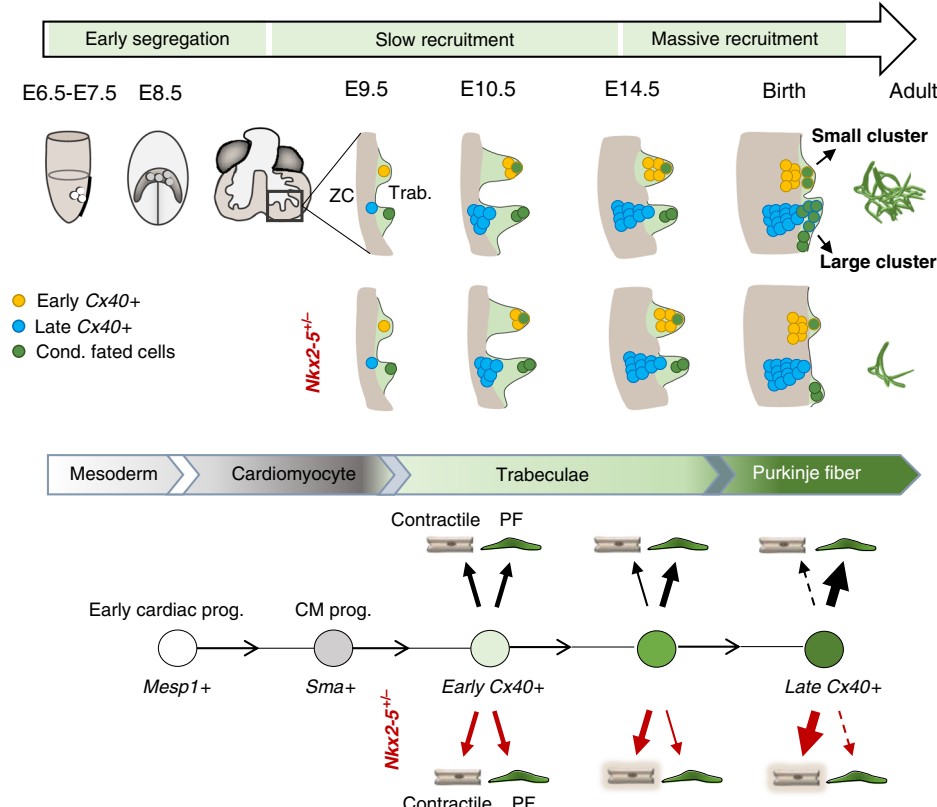

**Fig. 5 Model of the Purkinje fiber lineage segregation and temporal requirement of *Nkx2-5* during PF network morphogenesis.** Scheme showing the contribution of progenitor cells at different developmental stages and their contribution to the PF network. In the gastrulating embryo, *Mesp1*+ progenitors into the cardiomyocyte lineage (*Sma*+) contribute to ventricular myocardium. From E7.5, a subset of progenitors is already fated to the conductive lineage (green cells). During trabeculation (Trab.), *Cx40*+ (yellow) cells contribute to small PF clusters with reduced proliferation building a polyclonal PF network. These *Cx40*+ cells give rise to both conductive and working lineages and are unaffected by reduced *Nkx2-5* dosage. Cells from a highly proliferative progenitor subpopulation (blue) express Cx40 at later stages (from E10.5) as they are incorporated into trabeculae. These cells enter the conductive lineage at late fetal stage in a massive *Nkx2-5*-dependent recruitment to contribute to ellipsoidal structures of the PF network. In control mice the conductive potency of *Cx40*+ trabecular progenitors increases with time, whereas in *Nkx2-5+/−* mice, the conductive potency is progressively lost leading to very rare large clusters and a hypoplastic PF network. ZC compact myocardium.

for Nkx2-5 expression during development of distinct regions of the conduction system. Inductive cues that govern the VCS lineage fate choice are poorly understood. However, in *Nkx2-5* null embryos, the pool of conductive progenitors delineated by *Mink-LacZ*+ is absent[36] and deletion of *Nkx2-5* in embryonic trabeculae at E10.5 abolishes the formation of the PF network[21].

The presence of mixed conductive and working myocardial clones in our experiments shows that growth of the definitive peripheral VCS requires the progressive addition of cells from bipotent progenitor cells. Our lineage-tracing analysis supports a model by which lineage restriction to the conductive fate is progressive during heart morphogenesis[37]. While trabecular identity is conserved in *Nkx2-5+/−* cardiomyocytes, they progressively lose their capacity to differentiate into a PF fate. In wildtype hearts, bipotent cells with high proliferative capacity make a massive lineage contribution to the PF network before birth. Such progenitors have been described in the compact zone of the embryonic heart and contribute to ventricular myocardial expansion through the invasion of cells in the compact zone towards the inner surface of the ventricle during development[32]. Our data suggest that these progenitor cells are significantly more affected by *Nkx2-5* haploinsufficiency than early differentiating trabeculae. In contrast to previous data showing that myocardial *Nkx2-5* downregulates trabecular proliferation[20], we do not observe an increase of clone size in *Nkx2-5* heterozygote mice. Our results highlight the fact that Nkx2-5 levels play a major role

in the differentiation of trabecular cells towards a PF phenotype but not in their proliferation. This is consistent with previous studies suggesting that elevated levels of *Nkx2-5* expression defines a population of cardiomyocytes actively differentiating into a conductive phenotype and that precise temporal regulation of Nkx2-5 levels may be necessary for normal differentiation of PFs[38–40]. Conversely, reduced Nkx2-5 levels may play a role in the maintenance of proliferative status of progenitors. In accordance with this hypothesis, single-cell RNA-sequencing has demonstrated that *Nkx2-5+/−* mice exhibit higher numbers of immature cardiomyocytes in the left ventricle compared to control mice, especially at E14.5[41]. These data strongly suggest that PF differentiation is impaired in Nkx2-5 haploinsufficient cardiomyocytes that instead maintain a progenitor-like phenotype.

This study addresses clonal contributions to the developing PF network. The conditional deletion of one *Nkx2-5* allele at birth shows no effect on the differentiation of Cx40+ cells or morphogenesis of the PF network. Thus, our experiments demonstrate that a high level of Nkx2-5 is required during embryonic stages for the formation of the VCS rather than playing a role in post-natal maintenance of the PF network. Our temporal clonal analysis of trabecular cells demonstrates that the PF network grows principally by gradual addition of newly committed cells rather than by proliferation of conductive precursors. Indeed, the size of conductive clones labeled during embryogenesis is smaller than that of non-conductive clones, confirming that segregation into conductive fate

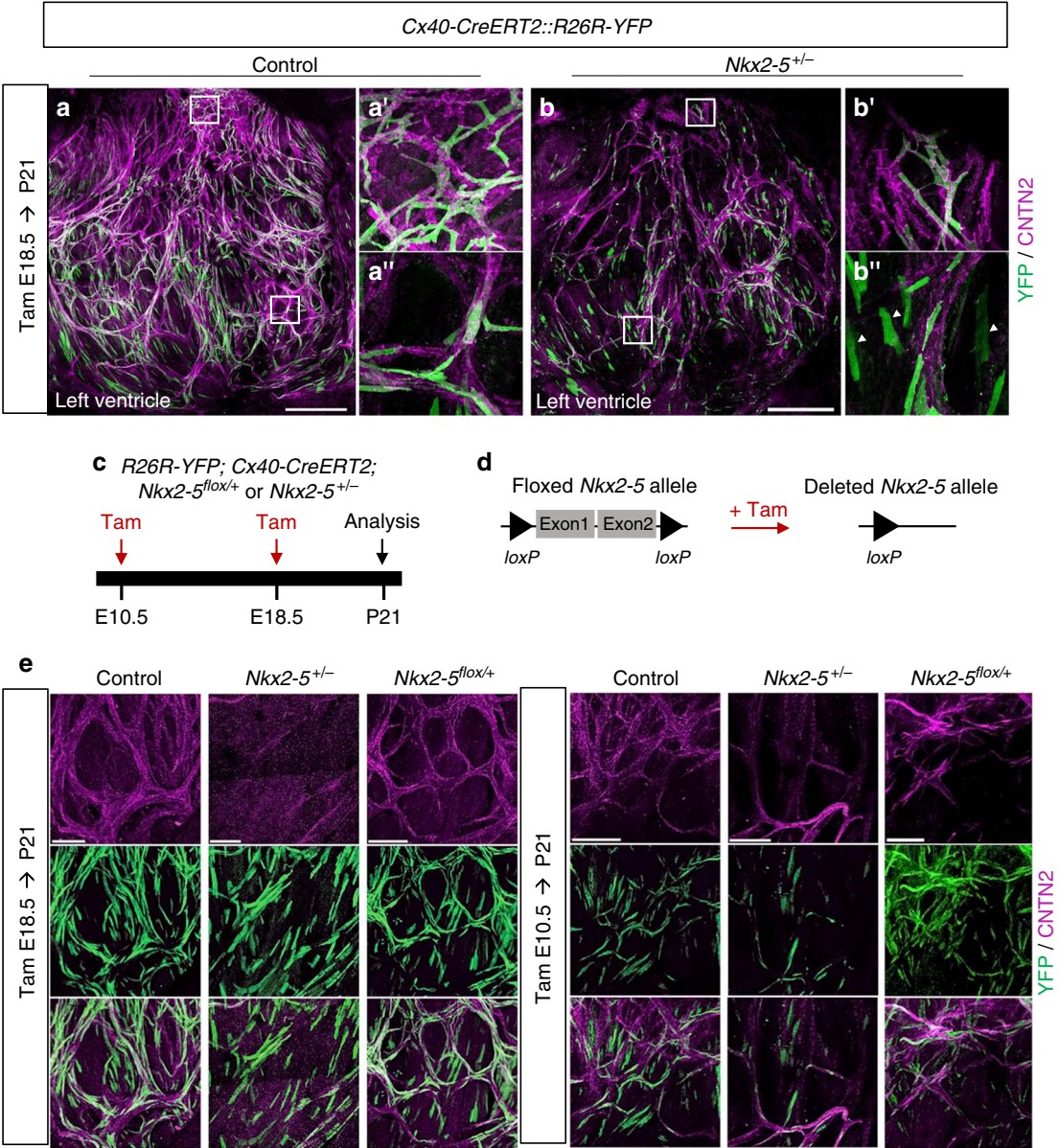

**Fig. 6 Cx40+ lineage tracing identifies that a maximal level of Nkx2-5 is required during embryonic stages for proper Purkinje network development.**
**a** and **b** Genetic tracing of Cx40+ cells labeled at E18.5 by Tam injection. Confocal images of whole-mount CNTN2-immunostaining of *Cx40-CreERT2::R26R-YFP* opened-left ventricle at P21. **a''** and **b''**, Immunostaining for YFP and CNTN2 on control, or *Nkx2-5⁺/⁻* mice shows E18.5 Cx40+-derived cells that participate to the PF network. Scale bars = 1 mm; **a'** and **b'** are high magnifications of left bundle branch; **a''** and **b''** are high magnifications of PF. **c–e** Strategy to induce conditional deletion of one copy of *Nkx2-5* allele (*Nkx2-5ᶠˡᵒˣ/⁺*) using *Cx40-CreERT2* line as inducible Cre driver. Deletion is performed at E18.5 when Cx40+ cells are mostly PF cells or E10.5 before the massive recruitment of late progenitors. *R26R-YFP* reporter line is used to compare the genetic tracing of Cx40+-derived cells within P21 control, *Nkx2-5⁺/⁻* and *Nkx2-5ᶠˡᵒˣ/⁺* hearts (*N* = 7 of each genotype for E18.5 and *N* = 3 for E10.5). **e** Confocal imaging of left PF network. Scale bars = 500 μm; arrowheads indicate non-conductive Cx40+-derived cells.

is associated with a loss of proliferative capacity[12]. The size of non-conductive clones generated after E14.5 labeling is similar to that of conductive clones, demonstrating the decrease of the proliferative capacities of cardiomyocytes at perinatal stages, as previously demonstrated[42]. As expected for a tissue with a low proliferative index[43], the stochastic labeling of conductive cells generated a mosaic pattern forming a polyclonal network with conductive cells sparsely distributed within the ellipses. The kinetics of apparition of conductive clones is biphasic with a first phase of recruitment from the heart tube stage until E14.5 during which the commitment of new conductive cells is slow and similar between *Nkx2-5⁺/⁻* and control mice. Maximal levels of Nkx2-5 is dispensable for the early phase of PF network formation including early commitment and

initial recruitment of conductive cells during trabecular development. We cannot exclude the possibility that other transcription factors are able to compensate for the reduced levels of Nkx2-5 during the early stages of development. Finally, these data suggest that the residual PF network present in *Nkx2-5⁺/⁻* hearts represents the persistence of this early network seen in hearts after the conditional deletion of one Nkx2-5 allele in early embryonic trabeculae. This structure defines an embryonic scaffold for the later PF network as, at late fetal stages, the number of new committed PF increased rapidly and are added to the early scaffold to form a complex PF network through a non-cell-autonomous mechanism dependent on maximal Nkx2-5 expression levels. The exact molecular mechanisms leading to the formation of this network need

further investigation. However, the resemblance of PF hypoplasia phenotypes in *Irx3* and *Etv1* mutants with that of *Nkx2-5* hetero-zygotes suggests a cooperative effect between these transcription factors[44,45]. Interestingly, the rapid recruitment of PF cells occurs during trabecular compaction which is also altered in *Nkx2-5*-mutant hearts and affected in human patients presenting non-compaction cardiomyopathies, often associated with arrhythmias[21,46]. Thus, impaired ventricular compaction may be responsible for the PF hypoplasia by suppressing late massive recruitment of newly differentiated PF.

In conclusion, our temporal clonal analysis of the mammalian VCS using the confetti system has identified sequential steps in the peripheral VCS development and the etiology of PF hypo-plasia observed in adult *Nkx2-5* heterozygous mice. Our obser-vation that progenitor-like cells persist in the mature *Nkx2-5*$^{+/-}$ heart is an important finding for the design of future therapeutic strategies. As *Nkx2-5* mutations are highly frequent and represent 4% of CHD, deciphering cues that may drive the fate of these cells into the conductive lineage would represent a valuable strategy to treat conductive disorders and ventricular arrhythmias.

## Methods

**Animals and injections**. The *Cx40-CreERT2*[23], *Sma-CreERT2*[47], *Mesp1-cre*[48], *R26R-YFP*[49], *R26R-Confetti*[31], *Nkx2-5-lacZ*[50], and *Nkx2-5-floxed*[51] mouse lines have been previously reported. Mice were housed at room temperature of $22 \pm 3$ °C, at a relative humidity between 45% and 65% and light:dark cycle of 12/12 h. For genetic tracing analysis, *Cx40-CreERT2* males were crossed with *R26R-YFP* females and tamoxifen was injected intraperitoneally to pregnant females (200 µl) for 1 or 2 consecutive days or to newborn in a single injection (10 µl). Tamoxifen (T5648, Sigma) was dissolved at the concentration of 20 mg ml$^{-1}$ in ethanol/sunflower oil (10:90). After tamoxifen treatment to pregnant females, newborn mice were recovered after caesarian and given for adoption to CD1 females. For prospective clonal analysis, *Sma-CreERT2* or *Cx40-CreERT2* males were crossed with *R26-Confetti* females and low doses of 4-Hydroxytamoxifen (4-OHT H6278, Sigma) was injected intraperitoneally to pregnant females in a single injection. 4-OHT was dissolved at the concentration of 10 mg ml$^{-1}$ in ethanol/Cremophor® EL (Sigma) solution (50:50). Before injection, 4-OHT was diluted in 1× PBS and 200 µl of this solution was injected intraperitoneally into pregnant females.

**Ethics statement**. All studies and procedures involving animals were in strict accordance with the recommendations of the European Community Directive (2010/63/UE) for the protection of vertebrate animals used for experimental and other scientific purposes. The project was specifically approved by the ethics committee of the IBDM SBEA and by the French Ministry of Research (APAFIS #01055.02). Husbandry, supply of animals, as well as maintenance and care of the animals in the Animal Facility of CNRS-IBDM (facility license #F 13 055 21) before and during experiments fully satisfied the animals' needs and welfare.

**Antibodies and immunofluorescence**. Antibodies used in this study are specific to sheep GFP (1:500; AbD Serotec, Bio-Rad, Hercules, CA, USA), chick GFP (1:500; 2010, Aves), and goat Contactin-2 (1:100; AF1714 R&D Systems, Min-neapolis, MN, USA), Donkey anti-goat-488 (1:500; A11055, Life Technologies), Donkey anti-goat-647 (1:250; A21447, Life technologies), Donkey anti-chick-488 (1:500; 703 545 155, Interchim), WGA-555 (1:500; Wheat germ agglutinin, Clin-isciences). For whole-mount immunofluorescence, hearts were dissected and fixed in 2% paraformaldehyde overnight at 4 °C, washed in PBS, permeabilized in PBS 1×/0.5% Triton X100 for 1 h and incubated for 3 h in saturation buffer (PBS 1×; 3% BSA; 0.1% Triton X100). The primary antibodies were incubated in saturation buffer for 2 days at 4 °C. Secondary antibodies coupled to fluorescent molecules were incubated in saturation buffer and after washes, hearts were observed under a Zeiss Lumar stereomicroscope V12 or Zeiss LSM780 confocal microscope. For immunofluorescence on cryostat sections, hearts were permeabilized for 20 min (PBS 1×/0.2% Triton X100) and blocked for 1 h (PBS 1×/3% bovine serum albu-min/0.1% Triton X100/0.05% Saponin). The primary antibodies were incubated in blocking buffer for overnight at 4 °C. Secondary antibodies coupled to fluorescent molecules were incubated in blocking buffer for 1 h. After washes hearts were observed under a Zeiss Z1 Apotome microscope. Imaging data were acquired using Zeiss software Zen Pro2012 Blue version 1.1.2.0 and processed using Adobe Photoshop 13.0.1; Image J 1.48v.

**Clonal analysis**. 4-OHT injections were performed at different time points of development to induce rare recombination events, generating well-separated clones. At 3 weeks (P21), hearts were dissected and pinned on Petri dish to expose the inner side of the left ventricle and fixed in 2% paraformaldehyde overnight at 4

°C and washed in PBS 1×. They were permeabilized for 1 h (PBS 1×/0.5% Triton X100) and blocked for 3 h (PBS 1×/3% bovine serum albumin/0.1% Triton X100). The primary anti-Contactin-2 antibody (1:200) was incubated in blocking buffer for 48 h at 4 °C. Secondary antibody coupled to fluorescent molecule, Donkey anti-goat-647 (1:500) was incubated in blocking buffer for 48 h and after washes, hearts were observed under a Zeiss LSM780 confocal microscope. Individual clusters were analyzed on virtual sections using the software Volocity 5.3.1. A single clone corresponds to a cluster composed of cells of only one color (RFP, GFP, YFP, or CFP) that are distributed in consecutive sections.

**Statistics and reproducibility**. Experiments were performed in biological and technical replicates as stated. Each experiment was repeated independently at least three times, with similar results. For each experiment, we used at least $n = 3$ animals, and technical replicates with at least $n = 3$ were used to calculate the statistical value of each analysis. Statistical analysis to compare differences between groups used ordinary one-way ANOVA test, unless otherwise stated. Statistics and graphs were generated using Prism-Graphpad 6.01.

**Reporting summary**. Further information on research design is available in the Nature Research Reporting Summary linked to this article.

## Data availability

All data supporting the findings of this study are available from the corresponding author on reasonable request.

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

## Acknowledgements

We are grateful to Sabrina Lefèvre-Beyer and Caroline Giessner for their involvement in the initiation and technical contributions to this project. We thank N. Bertrand, C. Cortes and F. Lescroart for critical discussions and reading of the manuscript. We acknowledge the France-BioImaging/PICsL infrastructure (ANR-10-INSB-04-01). This work was supported by the CNRS, the Association Française contre les Myopathies (AFM-Téléthon) (L.M.) and the Fondation pour la Recherche Médicale (FRM) (R.G.K.). C.C. was the recipient of GRRC/SFC Ph.D. and AFM-Téléthon postdoc fellowships.

## Author contributions

C.C., L.M. conceived the project and designed the experiments; C.C. performed most of the temporal clonal experiments; C.C. and L.M. performed genetic tracing experiments; C.C. performed the statistical analysis. R.G.K. and L.M. provided funding; R.G.K. reviewed and edited the manuscript; C.C. and L.M. wrote the manuscript.

## Competing interests

The authors declare no competing interests.
