## [Peer Review File · Nature Communications]

Reviewers' Comments:

Reviewer #1:

Remarks to the Author:

The manuscript by Choquet et al reports on the role of Nkx2-5 in the formation of the ventricular conduction system. Using genetic fate mapping and temporal clonal analysis, together with immunofluorescence techniques, the authors describe distinct phases in the embryological development of the ventricular conduction system. They show that in Nkx2-5 haploinsufficient mice embryos, the late phase of rapid recruitment of conductive precursors is abolished, resulting in an hypoplastic Purkinje network at birth. Moreover, they find an early segregation of cardiac progenitors organized in small clusters fated to conductive cells (prior to Cx40 expression) in an Nkx2-5 independent manner. Finally, they find that bipotent progenitors are still present at birth in haploinsufficient animals, which may open up therapeutic perspectives.

Overall, the study is of high quality and the manuscript well written. As noted by the authors, Nkx2-5 is a highly conserved transcription factor, whose mutations have been shown to be involved in congenital heart disease, associated with conduction disorders and arrhythmias. Although the methodology is highly specialized, the results have potentially important implications in the wider field of cardiology.

One major criticism is the lack of functional data. The authors show a significant reduction in the Purkinje network density in the haploinsufficient mice, but it is not clear to what extent this impacts cardiac electrical and contractile function. The functional impact may be mitigated by the redundancy in the distal network. Since the haploinsufficient animals appear to live beyond birth, it could be worthwhile to perform telemetry to quantify electrocardiographic parameters associated with ventricular conduction (such as QRS duration) and monitor for the occurrence of arrhythmias. Have these mice any signs of cardiac malformation (septal defects for example)?

The authors focus on the distal Purkinje network, but is there any indication that the more proximal parts of the ventricular conduction system (fascicles, His bundles, even AV node) are impacted by the Nkx2-5 insufficiency? This would presumably lead to a more severe phenotype.

The results obtained with the R26R-confetti mice are impressive. One question that arises is whether the spatial location of the different clusters is reproducible from one individual to another, or whether this occurs in a random manner. More generally, do the early/late and small/large clusters occur in anatomically distinct locations, for example with respect to the main fascicles, the His bundle, left vs right, base vs apex?

Figure 5 nicely illustrates the Purkinje fiber lineage segregation and indicates how and when Nkx2-5 is involved. However, it does not seem to take into account the slow and rapid recruitment stages of the Nkx2-5 dependent progenitors, as shown in Figure 2h. A related question is whether the Nkx2-5 independent progenitors also show different recruitment stages. Since Figure 5 serves as a general graphical scheme, it would be useful to include all aspects of Purkinje network formation.

Finally, some minor points:

- Mature Purkinje fiber cells are identified by the expression of Contactin 2, except for Figure 1. It was not clear to this reviewer why the same methodology was not applied here.
- Is it known whether functional heterogeneity exists across the distal Purkinje network (in terms of conduction, action potential, etc) and whether the Purkinje cells originating from the Nkx2-5 independent progenitors are functionally similar to the other Purkinje cells?
- Nkx2-5 also affects electrical and mechanical function of working cardiomyocytes (see for example D.J. Anderson Nature Comm 2018). The cardiac phenotype may therefore not solely be due to the defects in conduction system.

Reviewer #2:

Remarks to the Author:

Very well-prepared manuscript. I have no comments

Reviewer #3:

Remarks to the Author:

In human, the transcription factor NKX2-5 mutations represent around 4% of congenital heart diseases (CHDs) with a high prevalence of conduction disturbances and arrhythmias. As the authors point out, previous studies have highlighted the importance of Nkx2-5 for the formation and the maintenance of the conduction system. However, it is still unclear how NKX2-5 regulates formation of the complex PF network during ventricular morphogenesis. In this manuscript, Choquet C, et. al. use a rather elegant in vivo strategies to dissect the sequential events occurring during the development of the Purkinje fiber (PF) network by employing genetic fate mapping and temporal clonal analysis in mice. They identify cardiomyocytes committed to the PF lineage as early as E7.5. A polyclonal PF network emerges by progressive recruitment of conductive precursors to this scaffold from a pool of bipotent progenitors. At late fetal stages, the segregation of conductive cells increases during a phase of rapid recruitment to build the definitive PF network. But in Nkx2-5 haploinsufficient embryos, PF differentiation is impaired leading to failure to extend the scaffold and PF hypoplasia thus demonstrating a dose dependent function of Nkx2.5 on later stages of development. These results extend previous studies of these group to advance new insights into the etiology of PF hypoplasia.

Concerns and comments are listed below:

1. One of the major new contributions of the this work is observation that “maximal levels of Nkx2-5 is dispensable for early phase of PF network formation including early commitment and initial recruitment of conductive cells during trabecular development...but at late fetal stages, the number of new committed PF increased rapidly and are added to the early scaffold to form a complex PF network through a mechanism dependent on maximal Nkx2-5 expression levels” (Pg 13). However, the authors provide no evidence of what the mechanism(s) is for this ability to recruit PF cells in the later stages and how is this different from the mechanism of Nkx2-5 signaling in the early heart? It is a bit unsatisfying to not to have some evidence of the processes(s) that are dependent on Nkx2-5 signaling. We are bit more enlightened about what Nkx2.5 does, but have no more insight into how it does it. Is the effect of Nkx2.5 cell autonomous or perhaps non-cell autonomous? Further mechanistic insights would greatly enhance the contribution of these observations.
2. Because Cx40+ trabecular cells do not represent the totality of VCS progenitors and early Cx40-negative progenitor cells also contribute to the growth of more than half the PF network, how would Nkx2-5 regulate these Cx40-negative progenitor cells to develop into the PF network?
3. The authors should perhaps include a discussion of the recent work by Ren, et. Al., (Dev Cell 2019) suggesting that silencing of Nkx2.5 may be required for generation of pacemaker cells in the early embryo which would suggest generation of the PF system and pacemaker system are mechanistically as well as temporally and spatially distinct processes in conduction system development.

Minor Concerns:

1. Page 3 Line 72-73: This sentence is obviously talking about Supplementary Fig. 1, so should be “prior to analysis at postnatal day 21” instead of P7.

2. Page 4 Line 82-87: fig citations seem incorrect. The correct order should be "control hearts (Fig. 1d" (line 82), "wild-type situation (Fig. 1e)"(Line 84), "Nkx2-5+/- hearts (Fig. 1f)"? Because total PF cells contains more than half RFP+ cells, the title of Fig. 1f should be changed to "Number of RFP+ PF cells". It might be more reasonable to put Fig. 1f ahead of current Fig. 1e because Fig. 1e is based on your calculation of d & f.

3. Fig.1g: It might be better to make the arrow for the differentiation of embryonic trabecular progenitors into contractile cardiomyocytes thicker than that for conductive CMs (because only about 1/5 into VCS in wt) and make the arrow for conductive CMs in Nkx2-5+/- mice even thinner.

4. Fig.2c: only a singular arrow is needed for "Conductive" and "Non-conductive"? Line 7 in legend: "Arrows indicate cells positive---" should be "Arrowheads indicate cells positive---"?

5. Line 6-7 in legend of Fig.5: "This subpopulation of progenitors are Nkx2-5-independent." Because these cells still have a copy of Nkx-2-5 (+/-), how it could be true?

6. Re-organization of the Supplementary Fig citations would be quite helpful. The current order is confusing: SFig.1, SFig.2a, SFig.3, SFig.4a-c,d, SFig.2b, SFig.5, SFig.2c. In addition, Fig.2h is not cited. Actually, Fig.2h could be combined into Fig.5 to avoid repeating.

7. Page 9 Line 204: add citation as "trabecular cells at E14.5 (Fig.2g)"? Line 204-205 needs further explanation: How does this data "suggesting that PF derive entirely from these cells and not from another source of progenitors"?

8. Page 9 Line222: "their rate of proliferation" instead of "proliferative"?

9. Fig.6c-e: do you have any direct evidence to make sure efficient deletion in Nkx2-5-floxed/+ hearts by P1 tamoxifen injection? How many mice have been used? Because consistent and successful P1 tamoxifen injections for efficient deletion could be tricky and their dosage might be different from lineage tracing experiments.

10. In all quantification graphs, it would be easier to read if the authors were consistent by using the same color of bars for the same genotyping (i.e. always grey for controls and darker for Nkx2-5+/-).

REVIEWER COMMENTS

Reviewer #1 (Remarks to the Author):

The manuscript by Choquet et al reports on the role of Nkx2-5 in the formation of the ventricular conduction system. Using genetic fate mapping and temporal clonal analysis, together with immunofluorescence techniques, the authors describe distinct phases in the embryological development of the ventricular conduction system. They show that in Nkx2-5 haploinsufficient mice embryos, the late phase of rapid recruitment of conductive precursors is abolished, resulting in an hypoplastic Purkinje network at birth. Moreover, they find an early segregation of cardiac progenitors organized in small clusters fated to conductive cells (prior to Cx40 expression) in an Nkx2-5 independent manner. Finally, they find that bipotent progenitors are still present at birth in haploinsufficient animals, which may open up therapeutic perspectives.

Overall, the study is of high quality and the manuscript well written. As noted by the authors, Nkx2-5 is a highly conserved transcription factor, whose mutations have been shown to be involved in congenital heart disease, associated with conduction disorders and arrhythmias. Although the methodology is highly specialized, the results have potentially important implications in the wider field of cardiology.

One major criticism is the lack of functional data. The authors show a significant reduction in the Purkinje network density in the haploinsufficient mice, but it is not clear to what extent this impacts cardiac electrical and contractile function. The functional impact may be mitigated by the redundancy in the distal network. Since the haploinsufficient animals appear to live beyond birth, it could be worthwhile to perform telemetry to quantify electrocardiographic parameters associated with ventricular conduction (such as QRS duration) and monitor for the occurrence of arrhythmias. Have these mice any signs of cardiac malformation (septal defects for example)?

We agree with Reviewer 1 that it is important to determine the functional impact of PF hypoplasia. To this end, Nkx2-5 haploinsufficient mice have been previously studied by other groups and by us. In 2002, Tanaka et al. described the cardiac functional defects of Nkx2-5 haploinsufficient mice in great detail using transthoracic echo, ECG on anaesthetized animals, in vivo electrophysiology and patch-clamp experiments on ventricular cardiomyocytes (Tanaka et al., 2002). This study showed that Nkx2-5^{+/-} mice have AV conduction abnormalities, prolonged QRS intervals and inducible atrial and ventricular arrhythmias. At the cellular level, these authors described a decreased outward K⁺ current and prolonged action potential duration. In 2007, we showed that isolated Purkinje cells from Nkx2-5 haploinsufficient mice have identical action potential profiles to cells from control hearts and that the prolonged QRS duration observed in Nkx2-5^{+/-} mice results rather from ventricular conduction blocks and reduced conduction velocities, detected by optical mapping, due to PF hypoplasia (Meysen et al., 2007). Moreover, these mice present atrial septal defects with a reduced penetrance (20%) which depends on the genetic background (Biben et al., 2000; Tanaka et al., 2002). We didn't further analyze these features in our study as our work concentrated on ventricular Purkinje fiber network development. Echocardiography measurements did not reveal any differences in LV chamber dimensions in these mice (Tanaka et al., 2002).

In response to the reviewer's point, we have added two sentences in the introduction to better describe the functional and morphological defects of Nkx2-5^{+/-} mice.

Text in p 4:

"Nkx2-5^{+/-} mice reproduce the cardiac phenotype observed in patients, including atrial septal defects and atrioventricular (AV) conduction defects associated with hypoplastic development of the conduction system (Biben et al., 2000; Jay et al., 2004; Tanaka et al., 2002). Ventricular conduction defects such as prolonged QRS duration, conduction blocks and inducible ventricular arrhythmias have been correlated with a reduced PF network lacking ellipsoidal structures (Jay et al., 2004; Meysen et al., 2007; Tanaka et al., 2002)"

The authors focus on the distal Purkinje network, but is there any indication that the more proximal parts of the ventricular conduction system (fascicles, His bundles, even AV node) are impacted by the Nkx2-5 insufficiency? This would presumably lead to a more severe phenotype.

This paper focuses on the distal part of the VCS because these structures have not been analyzed in previous studies and because its morphogenesis is less known and still debated. In 2004, Jay et al. described progressive atrioventricular conduction defects associated with the loss of a specific region of AVN in Nkx2-5 heterozygous mutant mice (Jay et al., 2004). Moreover, the requirement of Nkx2-5 for the development and maintenance of the AVN has been confirmed by the absence of conductive progenitors of the ventral VCS in null embryos and its progressive loss in conditional null mice (Pashmforoush et al., 2004). However, in Nkx2-5 heterozygous mice, the His bundle and bundle branches are much less affected than

the AVN or peripheral PF demonstrating that the level of *Nkx2-5* expression is an important regulator of cardiomyocyte fate in specific subregions of the VCS (Meysen et al., 2007).

The results obtained with the R26R-confetti mice are impressive. One question that arises is whether the spatial location of the different clusters is reproducible from one individual to another, or whether this occurs in a random manner. More generally, do the early/late and small/large clusters occur in anatomically distinct locations, for example with respect to the main fascicles, the His bundle, left vs right, base vs apex?

We have mapped the position of each clone on a schematic representation of an opened LV and they are localized randomly over the septum surface in control and *Nkx2-5* mutant hearts (see revised supplementary Figure 2b). For the distribution of clones in the His-bundle we have previously shown that cardiac progenitors of the central conduction system segregate in the early embryo (Choquet et al., 2016).

Figure 5 nicely illustrates the Purkinje fiber lineage segregation and indicates how and when *Nkx2-5* is involved. However, it does not seem to take into account the slow and rapid recruitment stages of the *Nkx2-5* dependent progenitors, as shown in Figure 2h. A related question is whether the *Nkx2-5* independent progenitors also show different recruitment stages. Since Figure 5 serves as a general graphical scheme, it would be useful to include all aspects of Purkinje network formation.

We thank Reviewer 1 for this encouraging comment and we compiled these two schematic drawings in Figure 5 to create a general graphical scheme. The term *Nkx2-5* independent progenitors was improperly used and has been changed to: These *Cx40+* cells give rise to both conductive and working lineages and are unaffected by reduced *Nkx2-5* dosage.

Finally, some minor points:

- Mature Purkinje fiber cells are identified by the expression of Contactin 2, except for Figure 1. It was not clear to this reviewer why the same methodology was not applied here.

We agree with Reviewer 1 that we did not provide explanation for using *Cx40-RFP* as PF reporter in Figure 1 instead of *Contactin2*. We apologize for this but the size of the text being limited we considered that this omission is justified because we have data showing that both markers are expressed in the same cells (See Figure 1 below). When we initiated this study, we used the *Cx40-RFP* reporter to follow PF cells, however, we found that *Contactin2* antibodies work much better than RFP antibodies in wholemount and histological immunofluorescence. Moreover, as *Nkx2-5* is a regulator of *Cx40* expression, using *Contactin-2* in *Nkx2-5* mutants avoids any issues with reduced expression levels. We have checked for one heart that the number of *Cntn2+* and *RFP+* cells are identical on serial sections (see below), but as we had already quantified *YFP* and *Cx40-RFP* hearts ($n=4$ control and $n=4$ mutants), we didn't repeat this using *Cntn2* for all of them. We hope that these explanations will satisfy the reviewer's question.

Figure 1 : Whole-mount immunofluorescence staining of a *Cx40-GFP* heart (left) with a *cntn2* antibody (right) showing almost indistinguishable PF labeling (arrows).

	CNTN2+	CNTN2/YFP+	RFP	RFP/YFP+
E8 Apex R2G	201	82	266	87
E8 mil r2d	369	136	379	154

- Is it known whether functional heterogeneity exists across the distal Purkinje network (in terms of conduction, action potential, etc) and whether the Purkinje cells originating from the *Nkx2-5* independent progenitors are functionally similar to the other Purkinje cells?

This is an interesting remark from Reviewer 1 and we don't have the answer to this yet. We are planning to pursue our analysis of PF heterogeneity by combining transcriptomic and cellular electrophysiology in future experiments. However, our previous experiments showed that isolated PF from *Nkx2-5* haploinsufficient mice present normal AP parameters and *I_f* current (Meysen et al., 2007).

- *Nkx2-5* also affects electrical and mechanical function of working cardiomyocytes (see for example D.J. Anderson Nature Comm 2018). The cardiac phenotype may therefore not solely be due to the defects in conduction system.

We agree with Reviewer 1 that the cardiac phenotype of Nkx2-5 haploinsufficient mice results from defects of the entire myocardium including conductive and working cardiomyocytes. To discriminate between these two populations, we have generated the conditional knock-out of Nkx2-5 in the VCS by tamoxifen injection of Cx40-Cre::Nkx2-5^{flxed/flxed} newborn mice. Our unpublished results show that these mice present mild conduction defects (QRS with a normal duration but abnormal amplitude and shape) associated with mechanical dysynchrony and reduced ejection fraction. These data are not the subject of this manuscript focused on PF morphogenesis and will be presented in another publication. There is clear evidence in the literature that Nkx2-5 levels are higher in cells of the VCS compared to working cardiomyocytes (Thomas et al., 2001) supporting a critical role of maximal Nkx2-5 levels in the VCS for conduction system morphogenesis.

--

Reviewer #2 (Remarks to the Author):

Very well-prepared manuscript. I have no comments

We thank Reviewer 2 for this encouraging comment.

--

Reviewer #3 (Remarks to the Author):

In human, the transcription factor NKX2-5 mutations represent around 4% of congenital heart diseases (CHDs) with a high prevalence of conduction disturbances and arrhythmias. As the authors point out, previous studies have highlighted the importance of Nkx2-5 for the formation and the maintenance of the conduction system. However, it is still unclear how NKX2-5 regulates formation of the complex PF network during ventricular morphogenesis. In this manuscript, Choquet C, et. al. use a rather elegant in vivo strategies to dissect the sequential events occurring during the development of the Purkinje fiber (PF) network by employing genetic fate mapping and temporal clonal analysis in mice. They identify cardiomyocytes committed to the PF lineage as early as E7.5. A polyclonal PF network emerges by progressive recruitment of conductive precursors to this scaffold from a pool of bipotent progenitors. At late fetal stages, the segregation of conductive cells increases during a phase of rapid recruitment to build the definitive PF network. But in Nkx2-5 haploinsufficient embryos, PF differentiation is impaired leading to failure to extend the scaffold and PF hypoplasia thus demonstrating a dose dependent function of Nkx2.5 on later stages of development. These results extend previous studies of these group to advance new insights into the etiology of PF hypoplasia.

Concerns and comments are listed below:

1. One of the major new contributions of the this work is observation that “maximal levels of Nkx2-5 is dispensable for early phase of PF network formation including early commitment and initial recruitment of conductive cells during trabecular development...but at late fetal stages, the number of new committed PF increased rapidly and are added to the early scaffold to form a complex PF network through a mechanism dependent on maximal Nkx2-5 expression levels” (Pg 13). However, the authors provide no evidence of what the mechanism(s) is for this ability to recruit PF cells in the later stages and how is this different from the mechanism of Nkx2-5 signaling in the early heart? It is a bit unsatisfying to not to have some evidence of the processes(s) that are dependent on Nkx2-5 signaling. We are bit more enlightened about what Nkx2.5 does, but have no more insight into how it does it. Is the effect of Nkx2.5 cell autonomous or perhaps non-cell autonomous? Further mechanistic insights would greatly enhance the contribution of these observations.

To go deeper into the mechanistic understanding of the role of Nkx2-5 during Purkinje fibers recruitment, we have included new data on the conditional deletion of one Nkx2-5 allele during embryonic development in the revised manuscript (Figure 6). These data show that a large portion of the PF network are recruited after E10.5 in control hearts while the network is severely affected

when one copy of *Nkx2-5* is deleted at this embryonic stage. These results confirm that a maximum level of *Nkx2-5* is absolutely required for the phase of massive recruitment. In addition, because *Nkx2-5* deletion is only partial in these mutants as shown by the R26-YFP lineage, this demonstrates that wild-type cardiomyocytes are not recruited to form the PF network in these hearts. This confirms our previous analysis showing that the hypoplastic PF network was not rescued by wild-type cells in mouse chimeric mice (Meysen et al., 2007). Altogether these data suggest that the active recruitment phase at fetal stages occurs through a non-cell autonomous mechanism. However, our previous chimeric analysis points out that *Nkx2-5* is required cell-autonomously to induce PF differentiation (Meysen et al., 2007), suggesting that different mechanisms are required for differentiation of the PF and active recruitment of new PF during VCS morphogenesis.

2. Because Cx40+ trabecular cells do not represent the totality of VCS progenitors and early Cx40-negative progenitor cells also contribute to the growth of more than half the PF network, how would *Nkx2-5* regulate these Cx40-negative progenitor cells to develop into the PF network?

We consider that all cells of the peripheral VCS express Cx40 during their differentiation and that Cx40-lineage negative Purkinje fiber cells rather result from incomplete Cre recombination. However, we show that early progenitor cells of the PF network, prior to the time of activation of Cx40, do not require maximal Nkx2-5 levels. One possibility that will be explored in future single cell transcriptomic analysis of PF heterogeneity is to identify different signaling pathways from the transcriptomic profiles of trabecular cells during the early scaffolding and later recruitment phases.

3. The authors should perhaps include a discussion of the recent work by Ren, et. Al., (Dev Cell 2019) suggesting that silencing of *Nkx2.5* may be required for generation of pacemaker cells in the early embryo which would suggest generation of the PF system and pacemaker system are mechanistically as well as temporally and spatially distinct processes in conduction system development.

We discuss the strikingly different requirements for Nkx2-5 in PFs versus the pacemaker cells in the revised discussion. Comparison of our work with that of Ren et al might suggest that differential requirement for Nkx2-5 contributes to patterning the conduction axis in the developing heart.

Minor Concerns:

1. Page 3 Line 72-73: This sentence is obviously talking about Supplementary Fig. 1, so should be “prior to analysis at postnatal day 21” instead of P7.

We thank Reviewer 3 for this remark and we have changed the text.

2. Page 4 Line 82-87: fig citations seem incorrect. The correct order should be “control hearts (Fig. 1d)” (line 82), “wild-type situation (Fig. 1e)”(Line 84), “*Nkx2-5*+/- hearts (Fig. 1f)”? Because total PF cells contains more than half RFP+ cells, the title of Fig. 1f should be changed to “Number of RFP+ PF cells”. It might be more reasonable to put Fig. 1f ahead of current Fig. 1e because Fig. 1e is based on your calculation of d & f.

We have reorganized the text and the figure legend according to the Reviewer’s comment and started by the description of figure 1d before 1e and 1f. We did not change the order of 1f and 1e because the graph 1f results from calculation based on 1e data and is more logically placed after the graph in panel 1e.

3. Fig. 1g: It might be better to make the arrow for the differentiation of embryonic trabecular progenitors into contractile cardiomyocytes thicker than that for conductive CMs (because only about 1/5 into VCS in wt) and make the arrow for conductive CMs in *Nkx2-5*+/- mice even

thinner.

We have modified Figure 1g as suggested.

4. Fig.2c: only a singular arrow is needed for “Conductive” and “Non-conductive”? Line 7 in legend: “Arrows indicate cells positive---“ should be “Arrowheads indicate cells positive---“?

We prefer to keep two arrows to indicate the division of the parental cell. We have corrected “arrowheads”.

5. Line 6-7 in legend of Fig.5: “This subpopulation of progenitors are Nkx2-5-independent.” Because these cells still have a copy of Nkx-2-5 (+/-), how it could be true?

*We agree with Reviewer 3, we have rephrased this assumption to :
This subpopulation of progenitors is insensitive to reduced levels of Nkx2-5.*

6. Re-organization of the Supplementary Fig citations would be quite helpful. The current order is confusing: SFig.1, SFig.2a, SFig.3, SFig.4a-c,d, SFig.2b, SFig.5, SFig.2c. In addition, Fig.2h is not cited. Actually, Fig.2h could be combined into Fig.5 to avoid repeating.

We have reorganized the figures and supplementary figures in accordance to the Reviewer’s comment.

7. Page 9 Line 204: add citation as “trabecular cells at E14.5 (Fig.2g)”? Line 204-205 needs further explanation: How does this data “suggesting that PF derive entirely from these cells and not from another source of progenitors“?

We thank Reviewer 3 for this careful reading of the manuscript and agree that this assumption results from our overinterpretation of the data. While the percentages of Mesp1-derived and E14-Cx40-derived cells to the VCS are similar, these data result from different Cre activation and do not imply any correlation between these frequencies. So, we have deleted this sentence from the revised manuscript to avoid overinterpretation. This does not alter the rest of the results.

8. Page 9 Line222: “their rate of proliferation” instead of “proliferative“?

We have corrected this mistake in the revised manuscript.

9. Fig.6c-e: do you have any direct evidence to make sure efficient deletion in Nkx2-5-floxed/+ hearts by P1 tamoxifen injection? How many mice have been used? Because consistent and successful P1 tamoxifen injections for efficient deletion could be tricky and their dosage might be different from lineage tracing experiments.

We agree with Reviewer 3 that the frequency of deletion may differ from one reporter line to another. To validate our strategy we have compared the frequency of deletion of the Nkx2-5 floxed and YFP cassettes in homozygote hearts (Nkx2-5^{FL/FL}::RYFP^{FL/+}::Cx40^{Cre/+}) in which we can count the number of Nkx2-5-deleted cardiomyocytes (Nkx2-5-negative by IF) and the number of YFP-positive cardiomyocytes. From these experiments, we observed that all YFP-positive cells were deleted for Nkx2-5 demonstrating a good correspondence between these two reporters. Although, we did observed a number of Nkx2-5 deleted cardiomyocytes that did not express YFP. These results suggest that the Nkx2-5 Floxed cassette is more easily deleted than the YFP cassette and that YFP expression thus represents an underestimation of the Nkx2-5 deleted cells. Moreover, we have added new results showing a defective PF network when we deleted one copy of Nkx2-5 at an earlier time point, proving that the deletion is effective. We have used 7 control and 7 Nkx2-5^{FL/+} mice at E18.5 and 3 of each genotype at E10.5.

10. In all quantification graphs, it would be easier to read if the authors were consistent by using

the same color of bars for the same genotyping (i.e. always grey for controls and darker for Nkx2-5+/-).

We apologize for this inconstancy and we have changed the graphs in the revised manuscript to be homogenous.

References:

- Biben, C., Weber, R., Kesteven, S., Stanley, E., McDonald, L., Elliott, D.A., Barnett, L., Koentgen, F., Robb, L., Feneley, M., *et al.* Cardiac septal and valvular dysmorphogenesis in mice heterozygous for mutations in the homeobox gene Nkx2-5. (2000). *Circ Res* 87, 888-895.
- Choquet, C., Marcadet, L., Beyer, S., Kelly, R., and Miquerol, L. Segregation of Central Ventricular Conduction System Lineages in Early SMA+ Cardiomyocytes Occurs Prior to Heart Tube Formation. (2016). *J Cardiovasc Dev Dis* 3, 2.
- Jay, P.Y., Harris, B.S., Maguire, C.T., Buerger, A., Wakimoto, H., Tanaka, M., Kupersmidt, S., Roden, D.M., Schultheiss, T.M., O'Brien, T.X., *et al.* Nkx2-5 mutation causes anatomic hypoplasia of the cardiac conduction system. (2004). *J Clin Invest* 113, 1130-1137.
- Meysen, S., Marger, L., Hewett, K.W., Jarry-Guichard, T., Agarkova, I., Chauvin, J.P., Perriard, J.C., Izumo, S., Gourdie, R.G., Mangoni, M.E., *et al.* Nkx2.5 cell-autonomous gene function is required for the postnatal formation of the peripheral ventricular conduction system. (2007). *Dev Biol* 303, 740-753.
- Pashmforoush, M., Lu, J.T., Chen, H., Amand, T.S., Kondo, R., Pradervand, S., Evans, S.M., Clark, B., Feramisco, J.R., Giles, W., *et al.* Nkx2-5 pathways and congenital heart disease; loss of ventricular myocyte lineage specification leads to progressive cardiomyopathy and complete heart block. (2004). *Cell* 117, 373-386.
- Tanaka, M., Berul, C.I., Ishii, M., Jay, P.Y., Wakimoto, H., Douglas, P., Yamasaki, N., Kawamoto, T., Gehrman, J., Maguire, C.T., *et al.* A mouse model of congenital heart disease: cardiac arrhythmias and atrial septal defect caused by haploinsufficiency of the cardiac transcription factor Csx/Nkx2.5. (2002). *Cold Spring Harb Symp Quant Biol* 67, 317-325.
- Thomas, P.S., Kasahara, H., Edmonson, A.M., Izumo, S., Yacoub, M.H., Barton, P.J., and Gourdie, R.G. Elevated expression of Nkx-2.5 in developing myocardial conduction cells. (2001). *Anat Rec* 263, 307-313.

Reviewers' Comments:

Reviewer #1:

Remarks to the Author:

The authors have addressed my comments in a satisfactory manner. I have one final minor suggestion to expand the discussion on PF-related arrhythmia mechanisms. Given their previous work now cited in the introduction (Meysen et al), which shows a lack of electrophysiological changes at the cellular level in PF from Nkx2-5 +/- mice compared to wild type, it is unclear how Nkx2-5 might be involved in Purkinje-related VF trigger mechanisms mentioned in the first paragraph of the introduction.

Response to reviewers :

Reviewer #1 (Remarks to the Author):

The authors have addressed my comments in a satisfactory manner. I have one final minor suggestion to expand the discussion on PF-related arrhythmia mechanisms. Given their previous work now cited in the introduction (Meysen et al), which shows a lack of electrophysiological changes at the cellular level in PF from Nkx2-5 +/- mice compared to wild type, it is unclear how Nkx2-5 might be involved in Purkinje-related VF trigger mechanisms mentioned in the first paragraph of the introduction.

Indeed, our data emphasize the role of PF network architecture in cardiac activity and we hypothesize that architectural disturbances resulting from developmental defects of the PF network represent a high-risk factor for ventricular arrhythmias and SCD. To emphasize this idea we have added a sentence in the introduction:

While single PF cells from Nkx2-5 haploinsufficient mice display normal electrophysiological properties, ventricular conduction defects such as prolonged QRS duration, conduction blocks and inducible ventricular arrhythmias have been correlated with a reduced PF network lacking ellipsoidal structures¹⁶⁻¹⁸. Therefore, detailed understanding of how the PF network forms during development is required.